# Association between non-acute traumatic injury (TI) and heart rate variability (HRV) in adults: A systematic review protocol

Rabeea Maqsood[1]*, Ahmed Khattab[1], Alexander N. Bennett[2,3], Christopher J. Boos[1,4]

**1** Faculty of Health and Social Sciences, Bournemouth University, Bournemouth, United Kingdom, **2** Academic Department of Military Rehabilitation, Defence Medical Rehabilitation Centre, Stanford Hall, Loughborough, United Kingdom, **3** National Heart and Lung Institute, Faculty of Medicine, Imperial College London, London, United Kingdom, **4** Department of Cardiology, University Hospital Dorset, NHS Trust, Poole, United Kingdom

\* rmaqsood@bournemouth.ac.uk

**Data Availability Statement:** Once the systematic review is completed, it will be submitted to a peer-reviewed journal for publication and dissemination

## Abstract

Heart Rate Variability (HRV) is an indirect measure of autonomic function. Attenuated HRV is linked to worsening health outcomes including Major Adverse Cardiovascular Events (MACE). The relationship between traumatic injury (TI) and HRV has been limitedly studied. This research protocol has been designed to conduct a systematic review of the existing evidence on the association between non-acute TI and HRV in adults. Four electronic bibliographic databases (Web of Science, CINAHL, Medline, and Scopus) will be searched. The studies on non-acute (>7 days post injury) TI and HRV in adults will be included, followed by title-abstract screening by two reviewers independently. The quality and risk of bias of the included studies will be assessed using Axis and a six-item Risk of Bias Assessment tool for of Non-randomized Studies (RoBANS) respectively. Grading of Recommendations Assessment, Development and Evaluation (GRADE) will assess the quality of evidence. The extracted data will be synthesized using narrative syntheses and a Forest plot with or without meta-analysis- whichever permitted by the pooled data. This will be the first systematic review to examine the relationship between generalized TI and HRV in adults. (PROPSERO registration number: CRD: CRD42021298530) https://www.crd.york.ac.uk/prospero/display_record.php?ID=CRD42021298530.

## Introduction

Traumatic injury (TI) is a leading cause of death and long-term disability [1]. It is typically defined as any physical injury of sudden onset and severity requiring immediate medical care [2]. In total, 38 different injury mechanisms have been defined [3]. Among these road traffic accidents, falls and sport injuries tend be predominant among civilian populations [3] whereas some occupational population such as law enforcement officers have been reported to have four times higher risk of mortality than civilians as a result of traumatic injuries during assaults [4]. Similarly, in deployed military populations, there is a greater predominance of gunshots and blast, but this depends on the conflict and environment [5].

of findings. Wherever applicable, the data will be shared with the readers.

**Funding:** This project is a part of RM's Ph.D. studentship- jointly funded by Bournemouth University and the ADVANCE charity, UK.

**Competing interests:** The authors have declared that no competing interests exist.

Traumatic injury initiates an immediate autonomic response typified by an increase in sympathetic activation and gradual withdrawal of parasympathetic tone [6]. This autonomic response can be measured non-invasively using the concept of Heart Rate Variability (HRV)- which measures the fluctuations in cardiac inter-beat intervals in response to competing sympathetic nervous system and parasympathetic nervous system tones [6,7]. This can be examined in greater detail using time domain, frequency domain and non-linear methods, with each offering their own unique insight into autonomic balance [8].

Owing to HRV's ability to indicate somatic tissue damage before the onset of pain and the injury [9], HRV has also been proposed as a new vital sign and triage tool in trauma settings [10–12]. In addition to being an objective marker of autonomic function [13], HRV has also been shown to be a robust marker of injury severity and as a predictor of morbidity and mortality in patients with severe traumatic injuries to head, neck, torso, abdomen [14], burns [15], and Traumatic Brain Injury (TBIs) [16]. With this prognostic and diagnostic value, HRV has emerged as an increasingly useful measure of physical fitness and the identification of overtraining. Heart rate variability is a robust cardiovascular marker and reduced HRV has been strongly linked to major adverse cardiovascular events (MACE) and all-cause mortality [17].

In the last two decades, research into the impact of TI on HRV has been largely focused on acute trauma and during hospital admission [18–21]. While recent systematic reviews have summarized the relationship between HRV and depression [22], Post Traumatic Stress Disorder (PTSD) [23], and selected TBI [24] and spinal cord injury [25]; the current knowledge gap is in non-acute TI and HRV with the need to include multiple injury types. To date, no such systematic review has been undertaken. This systematic review is essential considering the recent conflicts in Iraq and Afghanistan which have brought the societal and long-term physical and psychological impact of TI among military conflicts into sharp focus. In a recent study of combat veterans, suffering previous TI in Afghanistan, it was shown that TI and its severity was independently associated with an increased cardiovascular risk [26]. Given the utility of HRV as a global non-invasive CVD risk its relationship to non-acute TI warrants further research interest.

This protocol for systematic review aims to address this research gap by reviewing the existing evidence base relating to generalized non-acute TI (excluding selected TBI, spinal cord, haemorrhage, PTSD and depression) [22–25] and HRV. This should lead to greater understanding of the longer-term impact of TI on autonomic function and the potential for preventive measures to moderate this pathway. The review question is: What is the association between non-acute traumatic injury and heart rate variability in adults with unselected TI versus uninjured controls? The objectives of this systematic review are to: examine the association between traumatic injury and heart rate variability, report which indices (whether time, frequency, non-linear) and injuries have been mostly reported in the literature and identify any patterns of confounders (age and time from injury) in relation to TI and HRV.

## Methods

This protocol has been designed using the Preferred Reporting Items for Systematic reviews and Meta-Analyses (PRISMA) protocol guidelines and checklist [27] (S1 Table). The protocol has been registered on PROSPERO- International Prospective Register of Systematic Reviews (CRD: CRD42021298530) and is available at: https://www.crd.york.ac.uk/prospero/display_record.php?RecordID=298530. The research question has been designed using the Population, Exposure, Comparator and Outcome (PECO) framework [28].

The results will be reported in the subsequent systematic review following the PRISMA 2020 guidelines [29]. Any changes to the protocol will be noted and explained there.

## Eligibility criteria

**Population.**   We aim to include studies with adult human participants aged 18 or above with no restriction on gender and occupation. The rationale is to include a broad spectrum of participants (civilian and military) [3–5]. Studies with participants who sustained chronic TI (>7 days post injury) and post initial hospital discharge will be included instead of acute injuries which led to death upon hospital admission. We selected the cut-off point of >7 days post injury based on a previous review on mild TBI [30] and to mitigate the impact of early physiological impact of acute trauma on HRV [18–21]. Any studies with animals or children and adolescents aged <18 years will be excluded.

**Exposure.**   For this systematic review, traumatic injuries are defined as any physical injuries or trauma such as injuries resulting from: burn, fall, explosion, blast, gunshot, amputation and accidents. Studies reporting traumatic brain injury (TBI), spinal cord injury, haemorrhage and psychological trauma such as PTSD, depression, and anxiety will be excluded [22–25].

**Comparator.**   Only studies with healthy controls (who didn't sustain any traumatic injuries) will be considered for inclusion.

**Outcome.**   Any measure of HRV- includes and not limited to: time domain (root mean square of successive differences between normal heartbeats (RMSSD), standard deviation of the average normal-to-normal (NN) intervals (SDANN), standard deviation of NN intervals (SDNN), frequency domain (low frequency power (LF), high frequency power (HF), total power (TP), LF/HF ratio) and non-linear measures for heart rate complexity (HRC) (e.g. entropy, SD1, SD2) will be considered as the primary outcomes in this study [8]. Studies measuring HRV either as a primary or secondary outcome will be eligible for inclusion. This study will include heart rate, type and mechanism of TI and time from TI as the secondary outcomes: There will be no restrictions on date, and geographical area. The inclusion and exclusion criteria are based on previous studies on the topic [3,4,18–25] and are listed in Table 1.

## Information sources

Cochrane library and PROSPERO registry were searched to see if a systematic review had already been undertaken. An initial scoping search was conducted to identify the relevant key terms in the study area and identify the gap. The preliminary search strategy has been developed around exposure and outcome by the authors with the help of an experienced librarian.

**Table 1. The inclusion and exclusion criteria.**

|  | Inclusion | Exclusion |
|---|---|---|
| **Population** | Young/middle/older adults from both sexes, aged >18, no history of cardiovascular disease | Animals, children, or adolescents aged <18, history of cardiovascular disease |
| **Exposure** | any physical trauma sustained (>7 days post injury) such as gunshot wounds, amputation, limb loss, burn and fall etc. | Studies which involve PTSD, depression, anxiety, head/brain injury, traumatic brain injury, spinal cord injury and haemorrhage. |
| **Comparison** | Controls with no traumatic injury. | No control |
| **Outcome** | Any index of Heart Rate Variability (HRV) | No HRV measure reported as a primary of secondary outcome. |
| **Setting** | Any | - |
| **Study Design** | Observational, cohort, cross-sectional, prospective, case control | Systematic reviews, reviews |
| **Language** | English | Other than English |
| **Date** | Any | - |
| **Publication status** | Published research papers | In-press, grey literature, conference proceedings, meeting abstracts, case reports, case series, opinion/editorial, in-vitro and animal study. |

The title and abstract fields of four electronic bibliographic databases (Medline, Web of Science, CINAHL and Scopus) will be searched with the search strategy (adapted for each database). MeSH terms and CINAHL subject headings will be added for "wounds and injury" and "heart rate variability" wherever applicable.

## Search strategy

Following is the master search strategy for Web of Science.

Exposure: trauma* OR wound* OR "blast" OR explosion* OR trauma N3 injur* OR "burn*".

Outcome: "heart rate variability" OR "HRV" OR "heart rate variation*" OR "heart rate complexity" OR "SDNN" OR "RMSSD" OR "autonomic function*" OR "autonomic reactivity" OR "HR-variability" OR "autonomic regulation" OR "autonomic activity".

The reference lists of included studies will also be scanned to supplement the searches and ensure the inclusion of important data sources which might have been missed in our search.

## Study records

**Data management.** The records will be exported to Mendeley Desktop (version 1.19.8) and maintained in Mendeley throughout the review. This will be followed by de-duplication which will be supplemented with manual screening and elimination of duplicates by both reviewers (RM and CJB).

**Selection process.** At the initial screening stage, the titles and abstract will be scanned by two reviewers (RM and CJB) independently. A supplementary selection criterion (S2 Table) will also be used at the initial stage to screen studies. Studies meeting the eligibility criteria will be retrieved for the full text. Full text articles will be evaluated by two independent reviewers (RM and CJB) using the inclusion and exclusion criteria. In cases of disagreement, a third reviewer (AK) will be invited to decide, followed by documenting the number of included and excluded studies. The search and screening results will be documented and displayed using the Preferred Reporting Items for Systematic Reviews and Meta-Analyses (PRISMA) flow diagram [29].

**Data collection process.** A data-extraction form (S3 Table) has been developed by RM and customized for this systematic review using the guidelines set by Centre for reviews and dissemination [31]. In case of absence, the data as reported by the authors in the studies will be extracted. The second reviewer (CJB) will cross-check the extraction performed by the lead reviewer (RM). The data-extraction form may also be continually developed as it will be piloted with two studies initially to test its reliability and comprehensiveness to extract data. Once completed, the data extraction forms will be cross reviewed by both reviewers (RM and CJB) to discuss discrepancies (if any). Any differences observed at this stage by the two reviewers will be resolved by inviting the third reviewer (AK). All data will be stored on Microsoft Excel and managed by the lead reviewer (RM). The corresponding authors will be contacted to obtain any missing data. If not available, the study will be excluded.

## Data items

For each article, following data (not limited to) will be extracted by the lead reviewer (RM): date of publication, authors, title, setting, country, study design, sample size, participants' gender and age, exposure and outcome variables, type of traumatic injury, injury-severity measure, time from injury, and measure of HRV, and heart rate. Wherever appropriate, means, standard deviation, p-values, correlation coefficients, confidence intervals and any other statistical findings will be extracted.

## Outcomes and prioritization

Any standardised index of HRV reported in the studies will be prioritised to be included as a primary outcome whereas heart rate will be reported as the secondary outcome. Injury-related characteristics such as time from injury and type of injury will be prioritised to be included in the narrative synthesis.

## Risk of bias in individual studies

Both reviewers (RM and CJB) will perform the critical appraisal of included studies using the Axis quality appraisal tool. The Axis checklist has 20 questions with response of either a yes (1) or a no (0) [32]. Study quality will be further categorized as low (<10), moderate (10–15) and high (>15) as previously used [33]. The final quality assessment score of each study will be documented and displayed on the quality appraisal table. Any disagreements at the quality assessment stage will be resolved by the third reviewer (AK).

Since non-randomized studies will be included in the review, the Risk of Bias (RoB) will be assessed using the six-item Risk of Bias Assessment tool for of Non-randomized Studies (RoBANS) [34]. The following six domains will be assessed in the included studies: the selection of participants, confounding variables, the measurement of exposure, the blinding of the outcome assessments, incomplete outcome data, and selective outcome reporting [34]. Studies scoring 0, 0–2, >2 will be rated as having low, moderate and high risk of bias, respectively as previously defined [33]. Two reviewers (RM and CJB) will independently evaluate the included studies for RoB against the RoBANS criteria. Disagreements will be resolved by discussion with the third reviewer (AK).

## Data synthesis

To detect and test the heterogeneity across studies, the Forest plot will be visually inspected followed by statistical tests- Chi-squared test and $I^2$ statistic [35] using Review Manager (RevMan 5.4.1) [36]. Outcome of the heterogeneity tests will inform the data-synthesis approach.

Given the methodological (different measurement tools/software used for ECG-recording and HRV analysis) and statistical heterogeneity (coefficient correlations, t-scores, odds ratio and means/medians etc), approach to data synthesis will be informed by the heterogeneity score. If heterogeneity is greater than 75%, it is likely that a narrative synthesis will be performed by tabulating the extracted data as suggested in the guidelines by Centre for reviews and dissemination [31] with a forest plot using Review Manager (RevMan 5.4.1) [36] without meta-analysis. However, if the characteristics of three or more studies are homogenous, a meta-analysis of the pooled data will be performed in RevMan using random-effects approach [37]. Mean difference (MD) or standardized mean difference (SMD) will be generated if the outcome of interest in the included study is given in the same or different outcome measures on a continuous scale [38], respectively.

## Meta-bias

For publication bias, the funnel plot will be visually inspected for asymmetry. If the number of studies included in the meta-analysis is less than ten, the Egger's test will not be used as suggested [39].

## Confidence in cumulative evidence

The quality of evidence will be assessed using Grading of Recommendations Assessment, Development and Evaluation (GRADE) by two reviewers (RM and CJB). GRADE is a system

to rank the quality of evidence in a systematic review and make recommendations [40]. Using the online tool, GRADEproGDT [41], each study will be ranked for quality of evidence into 4 ranks: very low, low, moderate, and high. GRADE evidence profile and summary of evidence will be reported in the subsequent systematic review.

## Discussion

Traumatic injury is one of the leading causes of death in adults aged <40 years [1]. The reported adverse health consequences of injuries sustained during recent conflicts in Afghanistan and Iraq have brought the impact of TI into greater clinical focus [26].

The results of several recent original publications have highlighted a plausible association between combat-related TI injury and subclinical cardiovascular risk [26,33,42]. The majority of this research has focussed on more well-established markers of cardiovascular risk such as heart rate, obesity, blood pressure, glucose and lipids [26]. The examination of the impact of TI on HRV has the potential to bridge the research gap. Heart rate variability is a unique non-invasive marker of autonomic balance that has been strongly linked to adverse health outcomes including MACE [17] across a broad spectrum of patient populations [14–16]. While the effects of acute TI on HRV have been studied [18–21], there appears to be a paucity of research on the longer terms impacts of non-acute TI and HRV, in particular beyond that of mild traumatic brain injury [24] and spinal cord injury [25]. Examination of the longer-term effects of TI on HRV has the potential to enhance existing research knowledge gaps and offer mechanistic insight to the reported elevation of cardiovascular risk with TI.

Addressing this research gap, this paper presents the protocol for the systematic review of literature on the association between non-acute TI and HRV. This systematic review is timely considering the recent trend of examining the long-term impact of TI in vulnerable groups such as military veterans.

This systematic review protocol offers several strengths. Firstly, it has been registered in the PROSPERO database to ensure a transparent conduct of the systematic review. Secondly, the literature searches conducted in the 4 databases will allow a comprehensive search. Thirdly, the rigour and the quality of the included studies will be assessed using validated critical appraisal and risk of bias tools along with GRADE, independently by two reviewers. Lastly, to our knowledge, this will be the protocol of the first systematic review which examines the association between a diverse spectrum of non-acute TIs and HRV in adults. Upon completion, the systematic review will be submitted to a peer-reviewed journal. However, some limitations are also anticipated such as heterogeneity in HRV data acquisition and analysis across the studies as found in the preliminary searches.

Nevertheless, the importance of this systematic review is not lessened by these limitations given the transparent conduct and rigorous evaluation of the included studies using validated tools. The outcomes of this systematic review may have implications to inform trauma care practice and intervention development for civilians and vulnerable populations such as military personnel and law enforcement officers.

## Supporting information

**S1 Table. This is the S1 Table PRISMA-P (Preferred Reporting Items for Systematic review and Meta-Analysis Protocols) 2015 checklist.**
(DOCX)

**S2 Table. This is the S2 Table the supplementary study selection criteria.**
(DOCX)

**S3 Table. This is the S3 Table data extraction form.**
(DOCX)

## Acknowledgments

We would like to acknowledge Mr. Caspian Dugdale, Academic Liaison Librarian at Bournemouth University, for helping us with building the search strategy.

## Author Contributions

**Conceptualization:** Rabeea Maqsood, Christopher J. Boos.

**Investigation:** Rabeea Maqsood.

**Methodology:** Rabeea Maqsood, Christopher J. Boos.

**Project administration:** Rabeea Maqsood.

**Resources:** Rabeea Maqsood.

**Software:** Rabeea Maqsood.

**Supervision:** Ahmed Khattab, Alexander N. Bennett, Christopher J. Boos.

**Validation:** Christopher J. Boos.

**Writing – original draft:** Rabeea Maqsood.

**Writing – review & editing:** Rabeea Maqsood, Ahmed Khattab, Alexander N. Bennett, Christopher J. Boos.

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
