## [Decision Letter · Decision Letter 0]

9 Jun 2022

PONE-D-22-12255Association between non-acute traumatic injury (TI) and heart rate variability (HRV) in adults: a systematic review protocolPLOS ONE

Dear Dr. Maqsood,

Thank you for submitting your manuscript to PLOS ONE. After careful consideration, we feel that it has merit but does not fully meet PLOS ONE’s publication criteria as it currently stands. Therefore, we invite you to submit a revised version of the manuscript that addresses the points raised during the review process.

ACADEMIC EDITOR:Dear authors,

The present study protocol is interesting but there still some improvements that can be made before further consideration.

For instance, some methodological aspects need more rational and support such as, the characteristics of the target population in study, the inclusion/exclusion criteria, and the development of some preliminar results and discussion to stretch the importance of such study.

Please consider the comments made by the reviewers.

Thank you==============================

We look forward to receiving your revised manuscript.

Kind regards,

Rafael Franco Soares Oliveira

Academic Editor

PLOS ONE

Journal Requirements:

2. hank you for stating the following in the Acknowledgments Section of your manuscript: 

"We would like to acknowledge Bournemouth University and the ADVANCE charity, UK for jointly funding the PhD studentship of RM at Bournemouth University"

"The authors received no specific funding for this work"

Additional Editor Comments:

Dear authors,

The present study protocol is interesting but there still some improvements that can be made before further consideration.

For instance, some methodological aspects need more rational and support such as, the characteristics of the target population in study, the inclusion/exclusion criteria, and the development of some preliminar results and discussion to stretch the importance of such study.

Please consider the comments made by the reviewers.

Thank you

Reviewers' comments:

Reviewer's Responses to Questions

**Comments to the Author**

1. Does the manuscript provide a valid rationale for the proposed study, with clearly identified and justified research questions?

Reviewer #1: Yes

Reviewer #2: Yes

2. Is the protocol technically sound and planned in a manner that will lead to a meaningful outcome and allow testing the stated hypotheses?

Reviewer #1: Partly

Reviewer #2: Yes

3. Is the methodology feasible and described in sufficient detail to allow the work to be replicable?

Reviewer #1: Yes

Reviewer #2: Yes

4. Have the authors described where all data underlying the findings will be made available when the study is complete?

Reviewer #1: Yes

Reviewer #2: Yes

5. Is the manuscript presented in an intelligible fashion and written in standard English?

Reviewer #1: Yes

Reviewer #2: Yes

6. Review Comments to the Author

You may also provide optional suggestions and comments to authors that they might find helpful in planning their study.

Reviewer #1: The present study is of interest to investigate the association between non-acute traumatic injury (TI) and heart rate variability (HRV) in adults, throughout a systematic review protocol.

Despite the interesting work, I strongly suggest following the comments to improve the quality of the manuscript.

1. Line 137 and 138 Population: "We aim to include studies with adult human participants aged 18 or above with no restriction on gender and occupation." Please explain why?

2. Line 139-140: "Studies with participants who sustained chronic TIs (>7 days post injury) and post initial hospital discharge will be included instead of acute injuries which led to death upon hospital admission." Please explain why?

3. Table1. Authors should explain why this inclusion/exclusion criterias were chosen across the text. Please use valid references to support your rationale.

4. Why authors decided to use RoB for the risk of bias analyses? Please clarify.

5. Despiste being this a "systematic review protocol" I think that will be very interesting to presente some "preliminar results" accompanied with some "preliminar discussion" (more robust) around the topic here presented. As it stands it is not clear how your questions/hypothesis will be really answered.

Reviewer #2: I carefully studied the file. The importance of the work in the introduction should be written thoroughly with up-to-date sources. Also it should be mentioned relevant studies in the introduction. The article is also in the discussion section Too short and too weak. It is not possible to be published in this way. It should be discussed with the articles included in the study. This section must be developed.

7. PLOS authors have the option to publish the peer review history of their article (what does this mean?). If published, this will include your full peer review and any attached files.

Reviewer #1: **Yes: **Júlio Alejandro Henriques da Costa

Reviewer #2: No

---

## [Author Response · Author response to Decision Letter 0]

17 Jun 2022

Also attached as the 'Response to reviewers'.

PONE-D-22-12255: Association between non-acute traumatic injury (TI) and heart rate variability (HRV) in adults: a systematic review protocol

We would like to thank the academic editor and the reviewers for providing insightful feedback on our manuscript which has improved our paper. The suggestions made by the academic editor and the reviewers have been incorporated into the manuscript and are highlighted. Please see below, in blue, for a point-by-point response to the reviewers’ comments. 

Academic Editor:

Dear authors,

The present study protocol is interesting but there still some improvements that can be made before further consideration. For instance, some methodological aspects need more rational and support such as, the characteristics of the target population in study, the inclusion/exclusion criteria, and the development of some preliminary results and discussion to stretch the importance of such study.

Please consider the comments made by the reviewers.

Response: Thank you for the feedback. Previously, for brevity, some methodological aspects were only briefly discussed. However, in the light of your helpful comments, the above suggestions have now been incorporated into our revised manuscript and are tracked. 

Reviewer #1: The present study is of interest to investigate the association between non-acute traumatic injury (TI) and heart rate variability (HRV) in adults, throughout a systematic review protocol.

Response: Thank you!

Despite the interesting work, I strongly suggest following the comments to improve the quality of the manuscript.

1. Line 137 and 138 Population: "We aim to include studies with adult human participants aged 18 or above with no restriction on gender and occupation." Please explain why?

Response: Thanks for this comment. This sentence is worded in this way to emphasise that our review is intended to be as inclusive as possible in terms of incorporating both men and women as well as a broad spectrum of participants (e.g. civilian and military). The following changes have been made (line 141-142).

“The rationale is to include a broad spectrum of participants (civilian and military) [3, 4, 5].”

Please note that the line numbers originally quoted by the reviewers have changed due to some addition in the revised manuscript. 

2. Line 139-140: "Studies with participants who sustained chronic TIs (>7 days post injury) and post initial hospital discharge will be included instead of acute injuries which led to death upon hospital admission." Please explain why?

Response: Apologies for the confusion. We agree that this needed more explanation. Setting the background, the following lines have now been added in the introduction section:

(Line 84-85): In the last two decades, research into the impact of TI on HRV has been largely focused on acute trauma and during hospital admission [14, 15, 16, 17]. 

(Line 88-90): (the current knowledge gap is in non-acute TI and HRV with the need to include multiple injury types. To date, no such systematic review has been undertaken)”.

-and under the heading “Eligibility criteria”:

(Line 145-147): We selected the cut-off point of >7 days post injury based on a previous review on mild TBI [29] and to mitigate the impact of early physiological impact of acute trauma on HRV [18, 19, 20, 21].

In sum, the rationale is that previous systematic reviews have looked at the relationship between HRV and acute trauma (selected TIs such as spinal cord, traumatic brain injury). To address this research gap, our systematic review will examine the association between non-acute (or generalised) TI and HRV in adults. 

Seven new references have been added (see below). Therefore, the in-text citations and reference list have been updated (tracked). 

• Batchinsky AI, Wolf SE, Molter N, Kuusela T, Jones JA, Moraru C, Boehme M, Williams K, Bielke P, Wade C, Holcomb JB, Cancio LC. Assessment of cardiovascular regulation after burns by nonlinear analysis of the electrocardiogram. J Burn Care Res. 2008;29(1):56-63.

• Colombo J, Shoemaker WC, Belzberg H, Hatzakis G, Fathizadeh P, Demetriades D. Noninvasive monitoring of the autonomic nervous system and hemodynamics of patients with blunt and penetrating trauma. J Trauma Acute Care Surg. 2008;65(6):1364-73.

• Riordan Jr WP, Norris PR, Jenkins JM, Morris Jr JA. Early loss of heart rate complexity predicts mortality regardless of mechanism, anatomic location, or severity of injury in 2178 trauma patients. J Surg Res. 2009;156(2):283-9.

• Mowery NT, Morris Jr JA, Jenkins JM, Ozdas A, Norris P. Core temperature variation is associated with heart rate variability independent of cardiac index: a study of 278 trauma patients. J Crit Care 2011;26(5):534-e9.

• Buker DB, Oyarce CC, Plaza RS. Effects of spinal cord injury in heart rate variability after acute and chronic exercise: a systematic review. Top Spinal Cord Inj. 2018;24(2):167-76.

• Krainin BM, Forsten RD, Kotwal RS, Lutz RH, Guskiewicz KM. Mild traumatic brain injury literature review and proposed changes to classification. J Spec Oper Med. 2011;11(3):38-47.

• Bhatnagar V, Richard E, Melcer T, Walker J, Galarneau M. Retrospective study of cardiovascular disease risk factors among a cohort of combat veterans with lower limb amputation. Vasc Health Risk Manag. 2019;15:409.

3. Table1. Authors should explain why this inclusion/exclusion criteria were chosen across the text. Please use valid references to support your rationale.

Response: Thank you for raising this. In our introduction, we have presented the background and rational of our target population as well as highlighting some of the key research gaps with reference to our targeted population with the inclusion of up-to-date references. These references have now been cited in line 165-166:

The inclusion and exclusion criteria are based on previous studies on the topic [3, 4, 18-25]

4. Why authors decided to use RoB for the risk of bias analyses? Please clarify.

Response: Thank you for highlighting this. We decided to use RoBANS for assessing RoB due to its validity and suitability for non-randomised and observational studies. The following lines (215-220) have now been added to the manuscript to clarify this point:

Since non-randomized studies will be included in the review, the Risk of Bias (RoB) will be assessed using the six-item Risk of Bias Assessment tool for of Non-randomized Studies (RoBANS) [33]. The following six domains will be assessed in the included studies: the selection of participants, confounding variables, the measurement of exposure, the blinding of the outcome assessments, incomplete outcome data, and selective outcome reporting [33]. 

5. Despite being this a "systematic review protocol" I think that will be very interesting to present some "preliminary results" accompanied with some "preliminary discussion" (more robust) around the topic here presented. As it stands it is not clear how your questions/hypothesis will be really answered.

Response: Thanks for this suggestion. Our preliminary searches indicate a rather small body of evidence on the topic. Sharing the preliminary results may be premature at the moment. However, the anticipated limitations of the included studies (e.g. heterogeneity in terms of HRV acquisition and analysis) have been discussed in “discussion”. Similarly, the anticipated data analysis approaches have been described under “data synthesis”. Furthermore, the importance of our work in the light of current research has also been discussed (please see our response to Reviewer 2’s 2nd comment). 

Reviewer #2: I carefully studied the file. The importance of the work in the introduction should be written thoroughly with up-to-date sources. Also, it should be mentioned relevant studies in the introduction. 

Response: Thanks for this suggestion. In the Introduction, the existing literature has been reviewed briefly from old to new studies. Addressing Reviewer 1’s comment as well, a few more studies have also been added to clarify the inclusion and exclusion criteria (Introduction section: line 84-90). References have now been cited in lines 99, 142, 146-147, 153, 166, 264). To our knowledge, all sources are up-to-date. 

The article is also in the discussion section Too short and too weak. It is not possible to be published in this way. It should be discussed with the articles included in the study. This section must be developed.

Response: Thank you for highlighting this. Considering Reviewer 1’s comment 5 as well, the discussion section has now been rewritten to include the importance, strengths, and limitation of the work in the light of current research as follows (Lines 258-295):

Traumatic injury is one of the leading causes of death in adults aged <40 years [1]. The reported adverse health consequences of injuries sustained during recent conflicts in Afghanistan and Iraq have brought the impact of TI into greater clinical focus [26]. 

The results of several recent original publications have highlighted a plausible associated between combat-related TI injury and subclinical cardiovascular risk [26, 32, 41]. The majority of this research has focussed on more well-established markers of cardiovascular risk such as heart rate, obesity, blood pressure, glucose and lipids [26]. The examination of the impact of TI on HRV has the potential to bridge the research gap. Heart rate variability is a unique non-invasive marker of autonomic balance that has been strongly linked to adverse health outcomes including MACE [17] across a broad spectrum of patient populations [14, 15, 16]. While the effects of acute TI on HRV have been studied [18, 19, 20, 21], there appears to be a paucity of research on the longer terms impacts of non-acute TI and HRV, in particular beyond that of mild traumatic brain injury [24] and spinal cord injury [25]. Examination of the longer-term effects of TI on HRV has the potential to enhance existing research knowledge gaps and offer mechanistic insight to the reported elevation of cardiovascular risk with TI.

Addressing this research gap, this paper presents the protocol for the systematic review of literature on the association between non-acute TIs and HRV. This systematic review is timely considering the recent trend of examining the long-term impact of TIs in vulnerable groups such as military veterans. 

This systematic review protocol offers several strengths. Firstly, it has been registered in the PROSPERO database to ensure a transparent conduct of the systematic review. Secondly, the literature searches conducted in the 4 databases will allow a comprehensive search. Thirdly, the rigour and the quality of the included studies will be assessed using validated critical appraisal and risk of bias tools along with GRADE, independently by two reviewers. Lastly, to our knowledge, this will be the protocol of the first systematic review which examines the association between a diverse spectrum of non-acute TIs and HRV in adults. Upon completion, the systematic review will be submitted to a peer-reviewed journal. However, some limitations are also anticipated such as heterogeneity in HRV data acquisition and analysis across the studies as found in the preliminary searches.

Nevertheless, the importance of this systematic review is not lessened by these limitations given the transparent conduct and rigorous evaluation of the included studies using validated tools. The outcomes of this systematic review may have implications to inform trauma care practice and intervention development for civilians and vulnerable populations such as military personnel and law enforcement officers.

Journal Requirements:

Response: Thank you for this comment. To best of our knowledge, the manuscript has been written and revised according to the PLoS one’s style requirements including those for file naming. Please see revised lines 432-434. We have also followed the Study Protocol template available at PloS One website:

https://storage.googleapis.com/plos-published-prod/c9fb/Study%20Protocol%20Article%20Template.pdf?X-Goog-Algorithm=GOOG4-RSA-SHA256&X-Goog-Credential=wombat-sa%40plos-prod.iam.gserviceaccount.com%2F20220615%2Fauto%2Fstorage%2Fgoog4_request&X-Goog-Date=20220615T101150Z&X-Goog-Expires=86400&X-Goog-SignedHeaders=host&X-Goog-Signature=1595c15be8ad346ebb2acbbb7688027e964e5dfcbb1f9663e245cdf425274c1fb484ea42eaef90dbe485c404f2643171bdfbaba483ade73457f8a32572567a0295bfc08be2a66590e6c3948e784138325fd232691b715db41f92f8bb7c80824e87a6e6fa75cfffcf6f8a904cda340cc1f0c83c4d4692fdad6266c41f1ef627e8795070419f341cf876874d35a91596e55a06fcbf90be420749757e434227a770dfdae4df567721362e6f0464b89e6b4de08810a34e34b38333440e4750d5ee79c98b4ad6d03963f56f85864bcc05310e0dcf6b2e0403217ecb2bc8221ee02211da4ba8a8e360aa8243460fcb1bec1cbbd335a31dbbc18463f0e38e7de290cdf6

"We would like to acknowledge Bournemouth University and the ADVANCE charity, UK for jointly funding the PhD studentship of RM at Bournemouth University"

"The authors received no specific funding for this work"

Response: Thank you for highlighting this. Please change the funding statement (line 32-33) to the following on my behalf:

This project is a part of RM’s Ph.D. studentship- jointly funded by Bournemouth University and the ADVANCE charity, UK. 

The Acknowledgment statement has now been revised (line 303-304)

We would like to acknowledge Mr. Caspian Dugdale, Academic Liaison Librarian at Bournemouth University, for helping us with building the search strategy.

---

## [Decision Letter · Decision Letter 1]

10 Jul 2022

PONE-D-22-12255R1Association between non-acute traumatic injury (TI) and heart rate variability (HRV) in adults: a systematic review protocolPLOS ONE

Dear Dr. Maqsood,

Thank you for submitting your manuscript to PLOS ONE. After careful consideration, we feel that it has merit but does not fully meet PLOS ONE’s publication criteria as it currently stands. Therefore, we invite you to submit a revised version of the manuscript that addresses the points raised during the review process.

We look forward to receiving your revised manuscript.

Kind regards,

Rafael Franco Soares Oliveira

Academic Editor

PLOS ONE

Journal Requirements:

Additional Editor Comments:

Dear authors,

The authors have addressed all comments of both reviewers. Nonetheless, reviewers 1 suggest that an explanation on the procedures to collect HRV should be added.

In addition, I found out that some methods need an update according to the most recent PRISMA 2020 guidelines, namely:

-the reference of the PRISMA 2020 update should be used;

-the PRISMA 2020 checklist should be followed in the exact same order, same sections and topics (as an example, assessment of methodological quality should be removed. Instead, the Study risk of bias assessment should be added).

Thank you

Reviewers' comments:

Reviewer's Responses to Questions

**Comments to the Author**

1. Does the manuscript provide a valid rationale for the proposed study, with clearly identified and justified research questions?

Reviewer #1: Yes

Reviewer #2: Yes

2. Is the protocol technically sound and planned in a manner that will lead to a meaningful outcome and allow testing the stated hypotheses?

Reviewer #1: Yes

Reviewer #2: Yes

3. Is the methodology feasible and described in sufficient detail to allow the work to be replicable?

Reviewer #1: Yes

Reviewer #2: Yes

4. Have the authors described where all data underlying the findings will be made available when the study is complete?

Reviewer #1: No

Reviewer #2: Yes

5. Is the manuscript presented in an intelligible fashion and written in standard English?

Reviewer #1: Yes

Reviewer #2: Yes

6. Review Comments to the Author

You may also provide optional suggestions and comments to authors that they might find helpful in planning their study.

Reviewer #1: The authors did a good job on reviewing the manuscript and answering all the revisions maded.

However, I think the paper could be more enriched if the authors could mention (briefly) how HRV can be measured in this type of population (civilian and military). We can use the same procedures?

Reviewer #2: I examined the Response "the Reviewer file". All corrections are made by the authors. Thank you so much for their effort. My decision is "accept"

7. PLOS authors have the option to publish the peer review history of their article (what does this mean?). If published, this will include your full peer review and any attached files.

Reviewer #1: **Yes: **Júlio Alejandro Henriques da Costa

Reviewer #2: **Yes: **Halil İbrahim Ceylan

---

## [Author Response · Author response to Decision Letter 1]

23 Jul 2022

The Response to Reviewers is also attached as a document. 

PONE-D-22-12255R1: Association between non-acute traumatic injury (TI) and heart rate variability (HRV) in adults: a systematic review protocol

We would like to thank the academic editor and the reviewers for providing insightful feedback on our manuscript once again. The suggestions made by the academic editor and the reviewers have been incorporated into the manuscript and are highlighted. Please see below, in blue, for a point-by-point response to the reviewers’ comments. 

Journal Requirements:

Please review your reference list to ensure that it is complete and correct. If you have cited papers that have been retracted, please include the rationale for doing so in the manuscript text, or remove these references and replace them with relevant current references. 

Response: Thank you for this comment. The reference list has now been proofread and revised according to the format available under PloS One’s reference section: https://www.nlm.nih.gov/bsd/uniform_requirements.html. 

Any changes to the reference list should be mentioned in the rebuttal letter that accompanies your revised manuscript.

Response: Thank you for highlighting this. Below are the reference numbers 27 and 29 which have now been updated in the manuscript for the PRISMA-P checklist 2015, and 2020 guidelines, respectively:

27. Moher D, Shamseer L, Clarke M, Ghersi D, Liberati A, Petticrew M, et al. Preferred reporting items for systematic review and meta-analysis protocols (PRISMA-P) 2015 statement. Syst Rev. 2015;4(1):1-9.

29. Page MJ, McKenzie JE, Bossuyt PM, Boutron I, Hoffmann TC, Mulrow CD, et al. The PRISMA 2020 statement: an updated guideline for reporting systematic reviews. Syst rev. 2021;10(1):1-1.

Owing to the addition of one new reference (29), the reference list has been updated accordingly (please see the tracked changes). Following are the updated reference numbers for which a few changes have been made in the reference list related to et al. and formatting: 

• 1, 2, 6,11, 13, 18, 26, 27, 29, 31, 34, 35, 37, 40, 41

Please note a few changes made in the manuscript as a result of proofreading:

• Line 69- ….which measures the fluctuations….

• Line 70-71: Deletion of abbreviations SNS, PNS

• Line 91-92- ..long-term physical and psychological impact…

• Line 110: addition of word ‘protocol’. 

• Line 116-117: The results will be reported in the subsequent systematic review following the PRISMA 2020 guidelines [29].

• Line 132: Deletion of abbreviation SCI 

• Line 151: Table 1: Deletion of repeated words history of 

• Line 425-426: S1 Table. This is the S1 table PRISMA-P (Preferred Reporting Items for Systematic review and Meta-Analysis Protocols) 2015 checklist.

If you need to cite a retracted article, indicate the article’s retracted status in the References list and also include a citation and full reference for the retraction notice.

Response: Thanks for this suggestion. I can confirm that a retracted article was not cited and was removed from the reference list. We do not aim to cite any retracted article. The manuscript attached contains the complete and revised reference list.

Additional Editor Comments:

Dear authors,

The authors have addressed all comments of both reviewers. Nonetheless, reviewer 1 suggests that an explanation of the procedures to collect HRV should be added.

Response: Thank you. Please see our response to reviewer 1’s comment.

In addition, I found out that some methods need an update according to the most recent PRISMA 2020 guidelines, namely:

-the reference of the PRISMA 2020 update should be used;

Response: Thank you for drawing our attention to this. However, we would like highlight that the PRISMA 2020 checklist refers to the reporting of a systematic review whereas the PRISMA-P 2015 checklist is used to report a systematic review protocol- as also suggested in the recent PRISMA 2020 guidelines:

Furthermore, PRISMA 2020 is not intended to inform the reporting of systematic review protocols, for which a separate statement is available (PRISMA for Protocols (PRISMA-P) 2015 statement 47,48). (Page et al. 2021).

Page MJ, McKenzie JE, Bossuyt PM, Boutron I, Hoffmann TC, Mulrow CD, et al. The PRISMA 2020 statement: an updated guideline for reporting systematic reviews. Syst rev. 2021;10(1):1-1.

While in this instance, we aim to keep the PRISMA-P checklist 2015 in the manuscript (systematic review protocol), the subsequent systematic review will be reported according to the recent PRISMA 2020 guidelines (Please see lines 116-117). 

-the PRISMA 2020 checklist should be followed in the exact same order, same sections and topics (as an example, the assessment of methodological quality should be removed. Instead, the Study risk of bias assessment should be added).

Thank you

Response: Thanks for raising this point. Since there is no significant difference in headings between the 2015 and 2020 checklists, the following section headings have now been renamed/updated (please see the tracked changes). The old sub-heading ‘assessment of heterogeneity’ has now been merged into ‘synthesis methods’ as per the order and headings of the PRISMA 2020 checklist:

• Information sources (line 153)

• Selection process (line 176)

• Data items (line 187)

• Study risk of bias assessment (line 206)

• Synthesis methods (line 227)

• Certainty assessment (line 244)

Please note that the section “Eligibility criteria” (line 119) has been moved up following the checklist order. 

Review Comments to the Author:

Reviewer #1: The authors did a good job of reviewing the manuscript and answering all the revisions made.

Response: Thank you. 

However, I think the paper could be more enriched if the authors could mention (briefly) how HRV can be measured in this type of population (civilian and military). We can use the same procedures?

Response: We appreciate your feedback. However, we would like to point out that this will be a systematic review and not a primary study. We do not aim to measure HRV in participants but present the results of previously reported work. However, we have cited references that provide a more in-depth review of HRV measurements and their interpretation. A detailed discussion on HRV measurement methods may be beyond the scope of this protocol. Nonetheless, we aim to address this in greater detail in the subsequent systematic review where it would be more relevant in the context of results. 

Reviewer #2: I examined the Response "the Reviewer file". All corrections are made by the authors. Thank you so much for their effort. My decision is "accept"

Response: Thank you!

---

## [Decision Letter · Decision Letter 2]

9 Aug 2022

PONE-D-22-12255R2Association between non-acute traumatic injury (TI) and heart rate variability (HRV) in adults: a systematic review protocolPLOS ONE

Dear Dr. Maqsood,

Thank you for submitting your manuscript to PLOS ONE. After careful consideration, we feel that it has merit but does not fully meet PLOS ONE’s publication criteria as it currently stands. Therefore, we invite you to submit a revised version of the manuscript that addresses the points raised during the review process.

ACADEMIC EDITOR:Dear authors,

Congratulations! Both reviewers already accept the work in the current form while I still have a minor revision to suggest.

Considering that this is a protocol for a systematic review that follows PRISMA-P 2015 (Moher, D., Shamseer, L., Clarke, M., Ghersi, D., Liberati, A., Petticrew, M., Shekelle, P., Stewart, L. A., & PRISMA-P Group (2015). Preferred reporting items for systematic review and meta-analysis protocols (PRISMA-P) 2015 statement. Systematic reviews, 4(1), 1. https://doi.org/10.1186/2046-4053-4-1) as you stated in the manuscript, I would like to suggest that authors also follow the same checklist item recommended in the reference.

For better clarity, after the section "Search strategy" you should present "Study records" in which you will include "data management" + "selection process" + "data collection process". After "Data items", it is suggested adding "Outcomes and

prioritization". "Study risk of bias assessment" should be replaced by "Risk of bias in individual studies". "Synthesis methods" should be replaced by "Data synthesis" as before. Then, you can add "Meta-bias(es)". Finally, instead of "Certainty assessment" used in the PRISMA 2020, it is suggest to replace by "Confidence in cumulative evidence".

I hope the authors can understand my suggestions.

After this step and if the authors follow these suggestions.

Thank you

We look forward to receiving your revised manuscript.

Kind regards,

Rafael Franco Soares Oliveira

Academic Editor

PLOS ONE

Journal Requirements:

Additional Editor Comments:

Dear authors,

Congratulations! Both reviewers already accept the work in the current form while I still have a minor revision to suggest.

Considering that this is a protocol for a systematic review that follows PRISMA-P 2015 (Moher, D., Shamseer, L., Clarke, M., Ghersi, D., Liberati, A., Petticrew, M., Shekelle, P., Stewart, L. A., & PRISMA-P Group (2015). Preferred reporting items for systematic review and meta-analysis protocols (PRISMA-P) 2015 statement. Systematic reviews, 4(1), 1. https://doi.org/10.1186/2046-4053-4-1) as you stated in the manuscript, I would like to suggest that authors also follow the same checklist item recommended in the reference.

For better clarity, after the section "Search strategy" you should present "Study records" in which you will include "data management" + "selection process" + "data collection process". After "Data items", it is suggested adding "Outcomes and

prioritization". "Study risk of bias assessment" should be replaced by "Risk of bias in individual studies". "Synthesis methods" should be replaced by "Data synthesis" as before. Then, you can add "Meta-bias(es)". Finally, instead of "Certainty assessment" used in the PRISMA 2020, it is suggest to replace by "Confidence in cumulative evidence".

I hope the authors can understand my suggestions.

After this step and if the authors follow these suggestions.

Thank you

Reviewers' comments:

Reviewer's Responses to Questions

**Comments to the Author**

1. Does the manuscript provide a valid rationale for the proposed study, with clearly identified and justified research questions?

Reviewer #1: Yes

Reviewer #2: Yes

2. Is the protocol technically sound and planned in a manner that will lead to a meaningful outcome and allow testing the stated hypotheses?

Reviewer #1: Yes

Reviewer #2: Yes

3. Is the methodology feasible and described in sufficient detail to allow the work to be replicable?

Reviewer #1: Yes

Reviewer #2: Yes

4. Have the authors described where all data underlying the findings will be made available when the study is complete?

Reviewer #1: Yes

Reviewer #2: Yes

5. Is the manuscript presented in an intelligible fashion and written in standard English?

Reviewer #1: Yes

Reviewer #2: Yes

6. Review Comments to the Author

You may also provide optional suggestions and comments to authors that they might find helpful in planning their study.

Reviewer #1: I am happy with the current version of the manuscript.

The authors did a good job on reviewing the manuscript and answering all the revisions maded.

Reviewer #2: All corrections were made by authours.. The article is improved by the authors.. I think that it's suitable for publication

7. PLOS authors have the option to publish the peer review history of their article (what does this mean?). If published, this will include your full peer review and any attached files.

Reviewer #1: **Yes: **Júlio Alejandro Henriques da Costa

Reviewer #2: No

---

## [Author Response · Author response to Decision Letter 2]

11 Aug 2022

Response to Reviewers is also attached as a separate file. 

PONE-D-22-12255R2: Association between non-acute traumatic injury (TI) and heart rate variability (HRV) in adults: a systematic review protocol

We would like to thank the academic editor and the reviewers for their feedback. The suggestions made by the academic editor have been incorporated into the manuscript and are highlighted. 

Please see below, in blue, for a point-by-point response to the reviewers’ comments. 

ACADEMIC EDITOR:

Dear authors,

Congratulations! Both reviewers already accept the work in the current form while I still have a minor revision to suggest.

Response: Thank you! 

Considering that this is a protocol for a systematic review that follows PRISMA-P 2015 (Moher, D., Shamseer, L., Clarke, M., Ghersi, D., Liberati, A., Petticrew, M., Shekelle, P., Stewart, L. A., & PRISMA-P Group (2015). Preferred reporting items for systematic review and meta-analysis protocols (PRISMA-P) 2015 statement. Systematic reviews, 4(1), 1. https://doi.org/10.1186/2046-4053-4-1) as you stated in the manuscript, I would like to suggest that authors also follow the same checklist item recommended in the reference.

For better clarity, after the section "Search strategy" you should present "Study records" in which you will include "data management" + "selection process" + "data collection process". After "Data items", it is suggested adding "Outcomes and

prioritization". "Study risk of bias assessment" should be replaced by "Risk of bias in individual studies". "Synthesis methods" should be replaced by "Data synthesis" as before. Then, you can add "Meta-bias(es)". Finally, instead of "Certainty assessment" used in the PRISMA 2020, it is suggest to replace by "Confidence in cumulative evidence".

I hope the authors can understand my suggestions.

After this step and if the authors follow these suggestions.

Thank you

Response: Thanks for raising this point. The revised manuscript now contains the suggested headings according to the PRISMA-P 2015 checklist. The following headings have been updated; line numbers are given in brackets.

Study records (174), Data management (175), Data collection process (190), Outcomes and prioritization (210), Risk of bias in individual studies (215), Data synthesis (233), Meta-bias (250), Confidence in cumulative evidence (254). 

Please note a few changes made in the manuscript as a result of change in the headings:

• Lines 176-179 moved into the section of Data management 

• Section on Outcomes and Prioritization added. 

• Lines 191-202 moved into Data Collection Process 

• Lines 251-253 moved into Meta-bias 

• Reference numbers updated in the reference list and the text [35-39]

---

## [Editor Report · Decision Letter 3]

15 Aug 2022

Association between non-acute traumatic injury (TI) and heart rate variability (HRV) in adults: a systematic review protocol

PONE-D-22-12255R3

Dear Dr. Maqsood,

We’re pleased to inform you that your manuscript has been judged scientifically suitable for publication and will be formally accepted for publication once it meets all outstanding technical requirements.

Kind regards,

Rafael Franco Soares Oliveira

Academic Editor

PLOS ONE

Additional Editor Comments (optional):

Dear authors,

Congratulations on your work.

My opinion is to accept.

Best regards
---

## [Editor Report · Acceptance letter]

18 Aug 2022

PONE-D-22-12255R3 

Association between non-acute traumatic injury (TI) and heart rate variability (HRV) in adults: a systematic review protocol 

Dear Dr. Maqsood:

I'm pleased to inform you that your manuscript has been deemed suitable for publication in PLOS ONE. Congratulations! Your manuscript is now with our production department. 

Kind regards, 

on behalf of

Dr. Rafael Franco Soares Oliveira 

Academic Editor

PLOS ONE